# Microbiota Gut–Brain Axis and Autism Spectrum Disorder: Mechanisms and Therapeutic Perspectives

**DOI:** 10.3390/nu17182984

**Published:** 2025-09-17

**Authors:** Andreas Petropoulos, Elisavet Stavropoulou, Christina Tsigalou, Eugenia Bezirtzoglou

**Affiliations:** Master Program in “Food, Nutrition and Microbiome”, Laboratory of Hygiene and Environmental Protection, Department of Medicine, Democritus University of Thrace, 68100 Alexandroupolis, Greece; elisabeth.stavropoulou@gmail.com (E.S.); xtsigalou@yahoo.gr (C.T.); empezirt@yahoo.gr (E.B.)

**Keywords:** autism spectrum disorder, microbiota gut–brain axis, dysbiosis, gut permeability, functional gastrointestinal disorders, probiotics, diet

## Abstract

**Background/Objectives**: Autism Spectrum Disorder (ASD) is a neurodevelopmental condition often accompanied by gastrointestinal (GI) symptoms and gut microbiota imbalances. The microbiota–gut–brain (MGB) axis is a bidirectional communication network linking gut microbes, the GI system, and the central nervous system (CNS). This narrative review explores the role of the MGB axis in ASD pathophysiology, focusing on communication pathways, neurodevelopmental implications, gut microbiota alteration, GI dysfunction, and emerging therapeutics. **Methods**: A narrative review methodology was employed. We searched major scientific databases including PubMed, Scopus, and Google Scholar for research on MGB axis mechanisms, gut microbiota composition in ASD, dysbiosis, leaky gut, immune activation, GI disorders, and intervention (probiotics, prebiotics, fecal microbiota transplantation (FMT), antibiotics and diet). Key findings from recent human, animal and in vitro studies were synthesized thematically, emphasizing mechanistic insights and therapeutic outcomes. Original references from the initial manuscript draft were retained and supplemented for comprehensiveness and accuracy. **Results**: The MGB axis involves neuroanatomical, neuroendocrine, immunological, and metabolic pathways that enable microbes to influence brain development and function. Individuals with ASD commonly exhibit gut dysbiosis characterized by reduced microbial diversity (notably lower *Bifidobacterium* and *Firmicutes*) and overpresentation of potentially pathogenic taxa (e.g., *Clostridia*, *Desulfovibrio*, *Enterobacteriaceae*). Dysbiosis is associated with increased intestinal permeability (“leaky gut”) and newly activated and altered microbial metabolite profiles, such as short-chain fatty acids (SCFAs) and lipopolysaccharides (LPSs). Functional gastrointestinal disorders (FGIDs) are prevalent in ASD, linking gut–brain axis dysfunction to behavioral severity. Therapeutically, probiotics and prebiotics can restore eubiosis, fortify the gut barrier, and reduce neuroinflammation, showing modest improvements in GI and behavioral symptoms. FMT and Microbiota Transfer Therapy (MTT) have yielded promising results in open label trials, improving GI function and some ASD behaviors. Antibiotic interventions (e.g., vancomycin) have been found to temporarily alleviate ASD symptoms associated with *Clostridiales* overgrowth, while nutritional strategies (high-fiber, gluten-free, or ketogenic diets) may modulate the microbiome and influence outcomes. **Conclusions**: Accumulating evidence implicates the MGB axis in ASD pathogenesis. Gut microbiota dysbiosis and the related GI pathology may exacerbate neurodevelopmental and behavioral symptoms via immune, endocrine and neural routes. Interventions targeting the gut ecosystem, through diet modification, probiotics, symbiotics, or microbiota transplants, offer therapeutic promise. However, heterogeneity in findings underscores the need for rigorous, large-scale studies to clarify causal relationships and evaluate long-term efficacy and safety. Understanding MGB axis mechanisms in ASD could pave the way for novel adjunctive treatments to improve the quality of life for individuals with ASD.

## 1. Introduction

Autism Spectrum Disorder (ASD) is a neurodevelopmental disorder characterized by deficits in social communication and interaction, alongside restricted or repetitive behaviors and interests [1]. Beyond its core behavioral symptoms, a high prevalence of gastrointestinal (GI) disturbances in ASD suggests a link between the gut environment and neurobehavioral manifestations [2]. The microbiota–gut–brain (MGB) axis represents a complex network of bidirectional communication between the gut microbiome, the GI tract, and the central nervous system (CNS) [3]. This axis includes neural (autonomic and enteric nervous systems, vagus nerve), endocrine (e.g., the hypothalamic–pituitary–adrenal (HPA) axis), immune, and metabolic pathways that enable gut microbes to influence brain function and vice versa [4]. Disruptions in the gut microbial population (dysbiosis) can affect the MGB axis, affecting brain development, behavior, and cognition [5].

Recent research highlights that children with ASD often harbor distinct gut microbiota profiles and metabolic signatures compared to neurotypical peers [6]. Such differences have been linked to increased intestinal permeability (“leaky gut”), immune dysregulation, and abnormal production of microbial metabolites that may influence the CNS [7]. Consequently, the gut microbiome has emerged as a potential factor in ASD etiology and symptom severity, as well as a target for novel interventions [8].

This narrative review provides a comprehensive overview of the MGB axis in the context of ASD, outlining key communication pathways and mechanisms, examining evidence of gut microbiota alteration in ASD, exploring associated GI pathology (dysbiosis, leaky gut, immune activation, functional GI disorders), and discussing current and prospective therapeutic approaches (probiotics, prebiotics, fecal microbiota transplantation, antibiotics, and nutritional interventions). By integrating findings across these domains, we aim to clarify how gut–brain interactions may contribute to ASD pathophysiology and identify how modulating the gut ecosystem could mitigate ASD-related symptoms.

## 2. Review Protocol

A narrative review approach was adopted due to the broad, interdisciplinary scope of the topic, encompassing microbiology, neuroscience, gastroenterology, and clinical psychology. A comprehensive literature search was conducted on PubMed, Web of Science, and Scopus databases, including publications up to April 2025. The searches included combinations of: “autism” or “ASD” with “microbiota gut–brain axis”, “gut microbiota”, “dysbiosis”, “intestinal permeability”, “leaky gut”, “short-chain fatty acids”, “gut–brain axis”, “probiotics”, “prebiotics”, “fecal microbiota transplant”, “antibiotics”, “diet”, and “functional gastrointestinal disorders” as key words. Boolean operators were used to refine the searches. The reference lists of the included articles were also screened to identify additional relevant studies.

The inclusion criteria were peer-reviewed English publications focusing on (i) the role of gut the microbiota or MGB axis in ASD or related new developmental outcomes, (ii) GI abnormalities and immune features in ASD, (iii) interventions targeting the gut microbiota in ASD or animal models of ASD, and (iv) relevant reviews or meta-analyses providing comprehensive backgrounds. Both clinical and epidemiological human studies, as well as mechanistic animal and in vitro studies were considered.

Relevant data were extracted and thematically categorized into six major topics: (1) MBX pathways and mechanisms, (2) the gut microbiota’s role in neurodevelopment and homeostasis, (3) microbiota composition differences in ASD, (4) dysbiosis and gut permeability in ASD, (5) functional GI disorders in ASD, and (6) therapeutic interventions targeting the MGB axis in ASD.

Given the narrative nature of this review, no formal quality appraisal of the included studies was conducted. However, studies with high levels of evidence (e.g., meta-analyses, controlled trials) were prioritized, and consensuses or recurring findings were emphasized. Seminal older studies were also included to provide historical context where appropriate.

## 3. The Microbiota Gut–Brain Axis: Pathways and Communication Mechanisms

The MGB axis is fundamentally a bidirectional communication pathway linking the gut and the brain through neurological, hormonal, immunological, and metabolic signals [3]. Neuroanatomical, neuroendocrine, immunological, metabolic, and microbial metabolite pathways are key components [9]. Neuroanatomical pathways facilitate rapid communication between the gut and the brain through direct connections, primarily via the vagus nerve and spinal pathways [10]. The central nervous system (CNS), through its autonomic outputs, can influence gut function, while visceral signals from the gut such as those triggered by the presence of nutrients-are relayed back to the brain [11]. The enteric nervous system (ENS), often referred to as “the second brain”, operates semi-autonomously but is modulated by both central inputs and local gut stimuli [12].

Neuroendocrine pathways play a central role in stress integration, primarily through the hypothalamic–pituitary–adrenal (HPA) axis [13]. The brain releases corticotropin-releasing hormone (CRH), which stimulates the adrenal glands to release cortisol, a stress hormone that can alter intestinal barrier function and modulate gut microbiota composition [14]. Gut microorganisms, in turn, produce metabolites that influence endocrine cells. For instance, enteroendocrine cells release hormones, like serotonin and ghrelin, in response to microbial signals. These hormones regulate appetite, mood, and gastrointestinal motility [15].

The microbiota and the gut mucosal immune system constantly communicate, maintaining immune tolerance under healthy conditions [16]. However, microbial imbalance (dysbiosis) can induce the production of pro-inflammatory cytokines including interleukin-6 (IL-6) and tumor necrosis factor-alpha (TNF-α). These cytokines may enter the bloodstream, cross the blood–brain barrier when elevated, and influence CNS immune cells like microglia [17]. Microglia are highly responsive to microbial signals, including components such as LPSs, and when they are overactive, they may trigger neuroinflammation [18].

Microbial products acting as signaling molecules are often used to communicate through metabolic and microbial metabolite pathways. Short chain fatty acids (SCFAs), such as acetate, propionate, and butyrate, are some of the most extensively studied. They are generated when dietary fibers are fermented. These SCFAs provide energy to colonocytes, strengthen gut barrier integrity, modulate immunological responses both locally and systemically, and even influence gene expression in the brain [19]. Tryptophan metabolites from gut bacteria or the host’s kynurenine pathway produce neuroactive compounds that alter serotonergic transmission, while gut-derived tryptophan is involved in processes that change neural circuits and influence brain function and behavior [20]. Additionally, gut microbes modify bile acids and synthesize important vitamins, notably B-vitamins, which have anti-inflammatory effects on the nervous system [21]. On the other hand, microbial LPSs, if allowed to translocate into circulation due to compromised gut permeability, can trigger systemic inflammation and activate microglia, contributing to neuroinflammatory processes [22].

The MGB axis links emotional and cognitive centers of the brain with peripheral intestinal function [23]. External factors such as stress can disturb this axis, by releasing cortisol, which affects gut permeability and alters the gut microbiota composition. Diet and antibiotic usage can also change the microbial community, leading to the transmission of abnormal signals back to the brain [4]. The MBG axis plays a crucial role in maintaining physiological homeostasis. Dysregulation of this system has been associated with a range of conditions, including anxiety, depression, and functional gastrointestinal disorders [24].

The MGB axis mainly communicates directly and indirectly [25]. The direct route involves gut-to-brain signaling via the vagus nerve and spinal afferents. For instance, some gut bacteria, such as *Lactobacillus rhamnosus*, can activate vagal pathways, which influence the release of neurotransmitters in the brain and change behavior. The indirect route operates through enteric nervous system (ENS) to CNS communication, mediated by intermediate endocrine or immune signals [26]. In this case, gut inflammation may not directly influence the brain, but the release of pro-inflammatory cytokines and stress-related hormones like corticotropin-releasing hormone (CRH) can elicit CNS responses, which may affect mood and behavior [27].

Researchers have identified that an imbalance in the microbes of the GI system may affect brain function. This is important for the study of ASD, which has both GI symptoms and neurodevelopmental dysfunction [28].

## 4. Role of Gut Microbiota in Neurodevelopment and Homeostasis

The gut microbiota starts to form during the prenatal period and plays a crucial role in shaping early brain development [29]. The mode of delivery significantly influences an infant’s initial microbial colonization; vaginal birth exposes the newborn to maternal vaginal and gut flora, such as *Lactobacillus*, whereas cesarean section results in colonization by skin and environmental microbes, often with lower initial microbial diversity [30]. The microbiota varies considerably during infancy. It starts with low diversity and gradually matures to an adult-like composition by around three years of age [30]. This microbial development coincides with critical windows of brain maturation. Disruptions to microbiota during this sensitive period, such as those caused by antibiotic exposure or formula feeding can have long-term consequences on both behavioral outcomes and immune system development [31]. One study demonstrated that the absence of a normal microbiota leads to exaggerated stress responses and social impairments, which can be partially reversed by introducing a conventional microbiota [32].

Microbes contribute to neurodevelopment by educating the immune system (establishing tolerance vs. inflammation) and producing metabolites that influence neuronal proliferation, BBB formation, and myelination [33]. Braniste et al. [34] demonstrated that the gut microbiota affects BBB permeability. Germ-free mice had a leakier BBB that normalized upon microbiota colonization. Also, microbial SCFAs in early life can induce microglial maturation and modulate neuroinflammation risk [34].

### 4.1. Homeostasis in Adulthood

In healthy states, the gut microbiota and host exist in mutualistic harmony. The microbiota improves digestion by breaking down complex polysaccharides, synthesizing vitamins (K, B12, folate), and training immune responses. The host, in return, provides nutrients and a stable environment. Eubiosis, the term for this equilibrium, is associated with normal neurochemicals levels and the integrity of the barrier [35].

The gut microbiome also modulates central neurotransmitter levels by changing the availability of their precursors, that is, the dietary and endogenous molecules required for neurotransmitter biosynthesis. For example, certain gut bacteria can produce GABA, an inhibitory neurotransmitter, or impact the levels of glutamate and serotonin in circulation [36]. A balanced microbiome supports normal levels of these neurotransmitters, whereas dysbiosis may skew the excitatory/inhibitory balance implicated in ASD [37]. Studies report that germ-free animals exhibit altered serotonin turnover and heightened HPA-axis activity (related to anxiety-like behavior), underscoring the microbiota’s role in stress regulation [19].

### 4.2. Microbiota in Immune Homeostasis

A healthy gut flora maintains a baseline of low-grade immune activation (sometimes called physiological inflammation) that fortifies gut barrier defenses without pathology [38]. Beneficial bacteria (like *Bifidobacterium* and *Lactobacillus*) induce anti-inflammatory molecules (IL-10, T regulatory cells) and enhance mucus production. This mitigates the risk of pathogenic invasion and excessive immune reactions. Notably disruptions to this balance have been connected to neurodevelopmental disorders [39]. Onore et al. highlighted immune dysfunction in ASD, including microglial activation and cytokine imbalances [40]. Chronic low-grade inflammation due to a dysbiotic microbiome could potentially influence brain development and behavior-a hypothesis that has been explored in ASD research [41].

### 4.3. Gut Microbiota as the “Second Brain”

The GI tract, often referred to as the “second brain” due to its extensive enteric nervous system, closely interacts with the gut microbiota to influence mood and behavior. For example, the microbiota can produce or stimulate the production of neurotransmitters. *Lactobacillus* and *Bifidobacterium* species can secrete GABA; gut microbes can convert dietary tryptophan to serotonin, which modulates mood [12].

The gut microbiota is integral to normal new developments and systemic homeostasis. It shapes immune and endocrine set-points, affects nutrient metabolism, and even modulates synaptic connectivity via neuroactive compounds. Disruption to this microbial signal during critical developmental periods could lead to lasting neurobehavioral changes, providing a plausible connection to conditions like ASD, where such processes are atypical [42].

## 5. Alterations in Gut Microbiota Composition in ASD

Numerous studies have compared the gut microbiota of individuals with ASD to that of neurotypical controls [43]. While results vary, certain patterns emerge. A consistent finding is reduced overall microbial diversity in ASD gut communities. This includes lower richness of beneficial bacteria (e.g., *Bifidobacterium*, some *Firmicutes* genera) and sometimes an overrepresentation of specific opportunistic microbes [44]. For instance, many ASD cohorts show *Bacteroidetes*/*Firmicutes* ratio imbalances and fewer Bifidobacteria, which are important for gut health [45]. Table 1 summarizes key studies on gut microbiota differences in ASD, highlighting the taxa reported as increased or decreased in ASD relative to controls.

Several genera stand out in ASD microbiomes. *Prevotella*, a fiber-fermenting genus abundant in neurotypical children, tends to be reduced in ASD, possibly due to restrictive diets low in fiber [8]. *Bifidobacterium*, crucial for fermenting milk oligosaccharides and maintaining gut barrier health, is frequently decreased in ASD. A low bacteria abundance may contribute to reduced SCFA production and diminished protection against pathogens [50]. Conversely, *Clostridiales* (which include *Clostridium* and *Desulfovibrio*) are often reported at higher levels [51]. Some *Clostridia* produce propionate and other metabolites that can affect behavior. In fact, Ossenkopp et al. showed that propionic acid administration in rats induced autism like behaviors [52]. *Desulfovibrio*, a sulfate-reducing bacterium, produces hydrogen sulfide, which can damage the gut epithelium and is more abundant in some ASD subjects [53].

Notably, *Sutterella* (a genus of unclear pathogenic potential) was detected in intestinal biopsies of a subset of children with ASD but not in controls. The significance of *Sutterella* is under investigation, but it may relate to gut mucosal immune interactions [53].

However, not all of these align perfectly. Some studies report increases in *Bacteoidetes* in ASD, while others report decreases; similarly, results were reported for *Firmicutes* [54]. This disparity may stem from factors like varied diets, ages, antibiotic exposures, or microbiomes in various parts of the world. It remains unclear whether changes in microbiota are a cause or consequence of ASD (or behaviors connected to it, such as a limited diet or anxiety that affects gut motility), underscoring the need for longitudinal and mechanistic studies to establish directionality [55].

Inconsistencies in reported microbiota alterations, such as differences in the *Bacteroidetes*/*Firmicutes* ratio, may stem from multiple factors. Geographic variation can influence microbial profiles through differences in environmental exposures and cultural dietary practices [56]. Dietary patterns, including macronutrient composition and food processing, are known to shape the gut microbiota and may differ substantially between study populations [57]. Methodological factors, such as DNA extraction protocols, sequencing platforms and bioinformatic pipelines can also affect taxonomic results [58]. In addition, microbiota composition changes naturally with age, and many studies include participants with a wide age range, which may confound results [59]. Medications exposure, particularly antibiotics, probiotics and psychotropic drugs, can alter microbial communities and is not consistently controlled [60]. In addition, perinatal and maternal influences such as mode of delivery, feeding type and maternal health conditions (e.g., stress, diet, gestational diabetes), play a role in early microbiota development [29].

## 6. Functional Implications of Dysbiosis in ASD

Regardless of precise taxonomic shifts, the collective picture is one of imbalance. Reduced diversity may result in an ecosystem less resilient to stressors, with fewer beneficial byproducts (e.g., SFCAs) and less competitive exclusion of pathogens [56]. Increased pathobionts could drive chronic low-grade inflammation or produce toxins. For example, certain *Clostridia* produce p-cresol, a toxin that can impair neurotransmitter systems, which has been found at higher levels in ASD urine [61].

Importantly, a study by Sharon et al. provided evidence of causality: when gut microbiota samples from human ASD donors were transplanted into germ-free mice, those mice developed behavioral traits analogous to ASD (repetitive behaviors, social deficits [32]. Germ-free mice that received microbiota from neurotypical donors, on the other hand, did not exhibit these behaviors. This suggests that certain characteristics of the ASD microbial community may contribute to features that are similar to those of ASD, although the specific microbes and mechanisms are still being investigated [62].

Future research with multi-omics (integrating metagenomics, metabolomics, and immunophenotyping) is needed to pinpoint how specific microbial changes translate to neurological impacts. Nonetheless, current evidence strongly supports that ASD is often accompanied by distinct gut microbiota alterations, forming a cornerstone of the MGB axis involvement in the disorder [6].

## 7. Dysbiosis, Leaky Gut, and Immune Activation in ASD

### 7.1. Dysbiosis and Intestinal Barrier Dysfunction

Dysbiosis refers to the disruption of the normal balance of the gut microbiota. In ASD, dysbiosis is characterized by a reduction in beneficial microorganisms and overrepresentation of opportunistic ones, as mentioned in [63]. One effect of dysbiosis is a compromised gut epithelial barrier. The term “leaky gut” describes an abnormal increase in intestinal permeability, which allows microorganisms and their products, including LPSs, to translocate into systemic circulation [7]. Evidence shows that leaky gut is more frequent in ASD: de Magistris et al. observed that about 37% of children with ASD had elevated gut permeability compared to 5% of controls [43]. D’Eufemia et al. also observed that 43% of children with ASD presenting GI symptoms had abnormal intestinal permeability [64]. These classic findings, together with a recent study by Teskey et al. that linked permeability markers to the severity of ASD behavior, suggest that some individuals with ASD may experience measurable alterations in their gut barrier integrity [65].

Mechanistically, dysbiosis (loss of butyrate producing bacteria, for instance) deprives colonocytes of nutrients like butyrate, which normally strengthen tight junctions between cells [65]. Pathobionts may produce metabolites that actively degrade mucus or open tight junctions. A dysbiotic gut often shows lower levels of *Akkermansia municiphila* (a mucus-degrading commensal that paradoxically promotes mucus thickness and integrity); ASD studies have noted reduced *Akkermansia*, potentially contributing to thinner mucus layers and leakiness [66].

Once the gut barrier is compromised, microbial components (LPSs, peptidoglycans, flagellin) enter the circulation and stimulate immune cells. LPSs in blood triggers toll-like receptor 4 (TLR4) on immune cells, causing the release of pro-inflammatory cytokines (IL-1β, TNF-α). Chronic elevation of these can lead to systemic inflammation and affect the brain [46]. Interestingly, ASD patients have been found to have higher plasma LPS levels correlating with more severe social impairment. Also, Emanuele et al. reported “low-grade endotoxemia” (slightly elevated blood LPS) in severe ASD cases, supporting this link [67].

### 7.2. Immune Activation and Neuroinflammation

ASD has been associated with immune dysregulation both peripherally and centrally. Many children with ASD show altered cytokine profiles, and postmortem studies reveal microglial activation and increased inflammatory mediators in the brain [68]. The gut microbiota likely plays a role in this neuroimmune crosstalk. For example, neuroinflammation in ASD could be partly driven by peripheral immune signals originating from the gut [69].

Microglia (CNS resident immune cells) are modulated by gut microbiota via circulating molecules [18]. SFCAs like butyrate, generally promote anti-inflammatory microglial phenotypes, whereas LPSs promote pro-inflammatory states [70]. Brown et al. [17] demonstrated that healthy microbiota protect against virus-induced neurologic damage through microglial TLR signaling. In ASD, a dysbiotic microbiota could skew microglial function toward a pro-inflammatory, synapse-pruning mode potentially affecting the neural circuits underlying behavior [17].

Molecular mimicry and antigenic stimulation by gut bacteria may also play a role in ASD. Some scientists suggest that specific gut bacteria cause immunological responses in genetically predisposed children and also affect neurological cells [71]. Despite similarities with other illnesses (such PANDAS, where a streptococcal infection causes neuropsychiatric symptoms) [72], there is not enough evidence to confirm that this is true for ASD.

### 7.3. Metabolic Endotoxemia and Brain Impact

When gut barrier integrity is compromised, gut-derived metabolites, such as propionic acid (PPA), a short-chain fatty acid produced by gut bacteria fermentation, can cross into the systemic circulation and enter the CNS [52]. In controlled trials, the administration of propionic acid in rodents, induced behavioral alterations such as repetitive behaviors, social isolation, and altered motor activity [73]. Similarly, p-cresol, derived from tyrosine fermentation by specific gut microorganisms, can inhibit dopamine beta-hydroxylase, which disrupts neurotransmitters balances. Some children with ASD were found to have elevated levels of p-cresol sulfate, particularly those with FGIDs [62]. These findings highlight how microbial metabolites, typically confined within the gut liver system, may enter the bloodstream and potentially modulate neurodevelopment and behavioral outcomes [74].

## 8. Leaky Gut and BBB

Interestingly, a similar mechanism may compromise the blood–brain barrier (BBB). Chronic peripheral inflammation and circulating toxins can loosen the BBB [75]. Animal ASD models show increased BBB permeability [76]. However, certain microbial products can also strengthen the BBB: acetate was shown to improve BBB tight junction integrity in mice [34]. Thus, not all microbial influence is detrimental; context and balance matter.

Negative influences may predominate in ASD: the combination of leaky gut, increased immune activation, and possible BBB leakiness creates a scenario for neuroinflammation [77]. Onore et al. emphasized the role of immune dysfunction in ASD pathophysiology, suggesting that immune activation and perhaps autoimmunity could contribute to neural symptoms. A gut-originating inflammatory cascade is a plausible contributor [40].

Dysbiosis in ASD is not a benign bystander, though it likely contributes to pathophysiological processes through leaky gut and heightened immune activation. The interaction between the gut microbiota and the immune and nervous systems may constitute a vicious cycle, where microbial dysbiosis promotes immune dysfunction and neuroinflammation, which in turn exacerbates behavioral and cognitive symptoms in ASD [78]. Therefore, therapeutic strategies aimed at restoring gut integrity and microbiota balance hold promise in potentially alleviating some aspects of ASD [79].

## 9. Functional Gastrointestinal Disorders and Their Link to ASD

FGIDs are gastrointestinal symptoms without detectable structural pathology. The reported prevalence of persistent gastrointestinal disorders in people with ASD ranges from 46% to 91%, but it is higher than in neurotypical children, suggesting that this is a major comorbidity [2]. Chronic constipation, diarrhea, abdominal discomfort, bloating, gastroesophageal reflux, and often changing bowel patterns are all common FGIDs in people with ASD. These symptoms match the Rome criteria for FGIDs, which include irritable bowel syndrome and functional constipation [80].

### 9.1. Brain–Gut Interaction in FGIDs

FGIDs are understood to involve disordered gut–brain communication. It is notable that children with ASD, who have intrinsic neural atypicality, also manifest disordered gut–brain interactions [36]. Luna et al. suggested that the signs of disrupted brain–gut interplay (like visceral hypersensitivity or irregular motility) are common in children with ASD and FGIDs [81]. Stress and anxiety, which are prevalent in ASD, can exacerbate FGIDs via HPA-axis activation and autonomic changes by reducing gut motility or pain thresholds [82]. Conversely, persistent gastrointestinal discomfort can worsen behavioral issues, possibly due to pain or microbiota-driven mood changes [83].

### 9.2. Specific FGIDs in ASD

Individuals with autism frequently report that constipation is the most common GI disorder. Potential contributing factors include restrictive, low-fiber diets, impaired gut motility potentially associated with autonomic nervous system dysfunction, adverse effects of certain medication, and alterations in gut microbiota composition [84]. Specifically, lower levels of beneficial microbes, such as *Bifidobacterium* and *Lactobacillus*, which support peristalsis, alongside an overabundance of methane producing microbes that slow intestinal transit, may contribute to the condition [85]. Chronic constipation can lead to complications such as fecal impaction and overflow diarrhea, which may necessitate diagnosis and management. Additionally, prolonged retention of fecal matter can enhance gut bacterial fermentation and increase microbial toxins absorption, which may, in turn, influence behavior and neurological function in individuals with ASD [86].

Individuals with ASD who experience chronic diarrhea often present with carbohydrate malabsorption and dysbiosis [87]. ASD has been linked to an overgrowth of some types of bacteria, namely, *Clostridiaceae* or *Sutterella*, which may alter stool consistency [88]. Additionally, deficiencies in digestive enzymes, particularly disaccharidases, have been reported in ASD, leading to incomplete carbohydrate digestion. This can result in osmotic diarrhea and further disrupt the gut microbial balance [89]. Diarrhea may also arise from rapid gastrointestinal transit, which can be triggered by heightened anxiety or autonomic nervous system imbalance [90]. Moreover, some children with ASD present with irritable bowel syndrome (IBS), characterized by alternating episodes of diarrhea and constipation, often influenced by psychological stress [91].

Pain and discomfort are common but often underrecognized gastrointestinal symptoms in individuals with ASD, largely due to communication challenges that make it difficult for many to express their discomfort verbally. Instead, such pain is often inferred from behaviors like pressing on the abdomen, curling up, or exhibiting irritability [92]. The sources of this discomfort can vary and may include bloating caused by dysbiosis and excessive gas production, pain related to constipation, or acid reflux symptoms. These issues can significantly impact quality of life and behavior, emphasizing the need for careful clinical observation and assessment [93].

One theory suggests that visceral hypersensitivity and FGIDs, like IBS, may be more pronounced in ASD due to an already sensitive nervous system. Neuroinflammation and altered pain processing in ASD might lower the threshold for gastrointestinal pain perception [83].

### 9.3. Microbiota-FGID Link

Dysbiosis in individuals with ASD may be a direct cause of FGIDs. One study linked lower levels of *Prevotella* and *Coprococcus* to severe GI symptoms. *Prevotella* helps break down fiber, suggesting it normally helps maintain comfortable gut function [8]. In the same way, increased *Clostridium* or *Desulfovibrio* growth may produce irritants affecting gut motility or secretion, leading to diarrhea or gastrointestinal discomfort [53].

Researchers have investigated whether GI disorder management can lead to the amelioration of ASD behavioral manifestations. Some modest studies and case reports show that relieving constipation or using gastrointestinal treatments can slightly ameliorate behavior, possibly by reducing discomfort or improving gut–brain signaling [81]. Petropoulos et al. noted that FGIDs in ASD are associated with abnormal levels of stress hormone profiles, pointing to physiological stress integration [2].

Chronic functional gastrointestinal disorders can significantly affect nutrient absorption in children with ASD, who often already face challenges due to selective or restrictive eating habits. Malabsorption caused by rapid intestinal transit or dysbiosis can cause deficiencies in important nutrients, especially fat-soluble vitamins and minerals [88]. These dietary deficiencies are especially concerning during critical periods of growth and brain development, as they may further impact neurological function and behavior [88]. This complicates the link between GI health and ASD highlighting the importance of early identification and management of dietary and GI issues [88].

Gastrointestinal disorders indirectly intensify symptoms of autism by contributing to emotional and behavioral challenges. Children with ASD who cannot effectively communicate their physical discomfort may express pain through increased irritability, aggression, or another behavioral outburst. These manifestations are often misinterpreted as purely behavioral, rather than somatic in origin [4]. In addition, GI disorders like acid reflux or abdominal discomfort can exacerbate sleep disturbances, which are already common in people with ASD leading to fatigue, reduced attention, and impaired daytime functioning [94]. Addressing gastrointestinal discomfort is essential not only for physical relief but also for supporting emotional regulation and overall quality of life in individuals with ASD [89].

FGIDs and ASD appear to share the same underlying causes, especially those related to neurotransmitter dysregulation. Serotonin, a major neurotransmitter that is very important for both mood and GI motility, is central to this connection. Most of the body’s serotonin is in the gut, which controls the motions and secretions of the intestines [23]. Individuals with ASD have been found to exhibit alterations in serotonin pathways, such as hyperserotonemia, which might be a sign of broader dysregulation of serotonin levels. Some researchers believe that dysregulated serotonin transmission in both the central and peripheral nervous systems may help explain why individuals with ASD present GI symptoms and behavioral problems at the same time [19].

FGISs are common in individuals with ASD and are strongly related to the MGB axis. They illustrate bidirectional distress: CNS abnormalities in individuals with ASD may increase susceptibility to FGIDs, while FGIDs may exacerbate CNS symptoms through microbiota and sensory pathways [7]. As a result, FGIDs management is increasingly seen as a component of holistic ASD care, with the potential dual benefit of improving GI comfort and possibly easing behavioral challenges.

## 10. Potential Therapeutic Approaches: Probiotics, Prebiotics, FMT, Antibiotics, and Nutrition

Given the evidence linking gut microbiota and MGB axis dysfunction to ASD, various therapeutic strategies have emerged aiming to modulate the gut ecosystem and, by extension, improve GI and behavioral outcomes. Therapeutic approaches used in microbiome dysbiosis include the use of probiotics, prebiotics, dietary interventions, antibiotics and FMT [95].

### 10.1. Probiotics

Probiotics are live beneficial microorganisms that, when administered in adequate amounts, confer health benefits [96]. The most commonly used genera are *Bifidobacterium* and *Lactobacillus* reflecting their natural prominence in a healthy gut. In ASD and related contexts, probiotics target dysbiosis and intestinal inflammation. The goal is often to restore *Bifidobacterium* levels given their noted deficits in ASD [97].

Probiotics exert a range of beneficial effects on gut and brain health through multiple interconnected mechanisms. They help strengthen gut barrier integrity by upregulating tight junction proteins and promoting mucus production, thereby reducing intestinal permeability. Probiotics also compete with pathogenic microbes, limiting their colonization while producing SCFAs and other metabolites that nourish enterocytes and modulate immune responses [98]. Additionally, certain probiotic strains may influence neurotransmitter activity. For example, *Lactobacillus* species produce gamma-aminobutyric acid (GABA), while Bifidobacteria can affect tryptophan metabolism, a precursor to serotonin [99]. These combined actions contribute to reduced systemic neuroinflammation, modulation of HPA axis overactivity, and potential increases in central levels of neuromodulatory compounds, such as brain-derived neurotrophic factor (BDNF), as observed in some animal studies. Through these pathways, probiotics may help regulate both gastrointestinal and neural behavioral symptoms, particularly in populations such as individuals with ASD [100].

Several clinical trials and case studies have explored the potential benefits of probiotics in children with ASD, focusing on both gastrointestinal and behavioral outcomes. Grossi et al. [53] presented a case series in which long-term administration of high-dose multi-strain probiotics led to notable improvements in core ASD symptoms including enhanced social interaction and communication. These improvements were accompanied by normalized gut microbial composition and reduced tumor necrosis factor (TNF) levels, suggesting decreased systemic inflammation [53]. In another important study, *Lactobacillus plantarum* was administered to children with ASD over a three-month period, resulting in improved stool consistency and modest behavioral gains, although not all trials have demonstrated consistent behavioral effects [101]. Navarro et al. reviewed probiotic use in ASD and found frequent improvements in gastrointestinal symptoms, with some anecdotal reports of enhanced eye contact and mood; however, they concluded that the evidence remains preliminary and inconclusive [102]. Nonetheless, these findings are limited by small sample sizes and the absence of placebo controls, highlighting the need for larger, well-controlled trials to confirm the therapeutic potential of probiotics in ASD.

Animal studies provide biological plausibility: Hsiao et al. [103] showed that the probiotic *Bacteroides* fragilis corrected leaky gut and improved autism-like behaviors in a mouse model. In mice, *Lactobacillus rhamnosus* reduced anxiety and depressive behaviors by lowering corticosterone [104]. In piglets, *Bifidobacterium* supplementation increased hippocampal metabolites related to neurodevelopment [41].

Probiotics are generally safe and well tolerated in children, though caution is warranted in immunocompromised individuals. Mild side effects can include transient gas or bloating. Some children with ASD and severe allergies or gut inflammation, may experience initial irritability as the microbiome shifts [105].

Overall, probiotics appear promising; however, the outcomes of several studies regarding behavioral changes are mixed, despite the apparent GI benefits. Larger placebo-controlled triads are needed [106]. Anecdotal positive outcomes and the known gut health improvements, make probiotics and compelling complementary therapy for many families affected by ASD. However, it is important to note that the current evidence base for probiotic use in ASD remains limited, with heterogenous study designs and small sample sizes impeding firm conclusions [103].

### 10.2. Prebiotics

Prebiotics are non-digestible food components, including fibers like fructo-oligosaccharides or galacto-oligosaccharides. They selectively stimulate the growth or activity of beneficial gut bacteria. Unlike probiotics, which introduce microbes, prebiotics nourish existing favorable bacteria [107].

Prebiotics play an important role in the MGB axis because they are fermentable substrates for beneficial gut bacteria, leading to the production of SCFAs [108]. SCFAs exert a wide range of beneficial effects such as reduced colon pH, gut barrier maintenance and the proper function of the MGB axis [109]. Although fewer studies have specifically examined prebiotics alone in the context of ASD, emerging evidence suggests promising effects on gut health and associated symptoms. Guarino et al. [109] reported that prebiotics promote the growth of beneficial bacteria genera such as *Bifidobacterium* and *Lactobacillus*, which are often found in reduced abundance in individuals with ASD. By lowering gut pH and inhibiting the production of harmful metabolites, prebiotics may help alleviate gastrointestinal symptoms commonly seen in this population [109]. An in vitro study referenced by Fattorusso et al. [110] further supports this potential, showing that probiotic supplementation can shift the gut microbiota toward a healthier composition, which may in turn reduce GI discomfort [110]. Adding prebiotics such as inulin or galacto-oligosaccharides to the diets of children with ASD has been linked to better stool consistency and less bloating in the clinic [111]. There is still not enough systematic clinical research, and more controlled studies are needed to adequately prove the effectiveness and therapeutic usefulness of prebiotics in treating GI and behavioral symptoms connected to ASD.

A concept gaining traction is microbiota-accessible carbohydrates (MACs): Diets low in MACs, such as highly processed diets, may starve gut microbes. Many children with ASD and restricted diets may inadvertently consume fewer MACs, contributing to dysbiosis. Prebiotic supplements can help counteract this, by providing fermentable fibers [109].

### 10.3. Symbiotics

Symbiotics are a combination of probiotics and prebiotics that offer a synergistic approach to enhancing gut health by simultaneously introducing beneficial bacteria and providing the nutrients they need to thrive. Prebiotics serve as a selective food source for specific probiotic strains, promoting their growth and activity within the gut [112]. By pairing these components, symbiotics can more effectively support the establishment and persistence of beneficial microbes compared to probiotics or prebiotics alone. This enhanced colonization may lead to greater improvements in gut barrier function, immune modulation, and possibly neurobehavioral outcomes, particularly in populations with disrupted microbiota such as individuals with ASD [113].

### 10.4. Fecal Microbiota Transplantation (FMT)

FMT is the process of transferring stool (and the whole microbial population) from a healthy human donor to the GI tract of a recipient. It is a radical way to change the gut microbiota [114]. To address safety concerns, FMT human donor screening involves a detailed medical and lifestyle history, laboratory testing of blood and stool for infectious agents and quarantine measures. Repeat testing is performed for regular donors. These steps are consistent with recent guidelines that include precautions for emerging pathogens [115].

If dysbiosis plays a major role in ASD symptom manifestations, then restoring a more balanced, neurotypical-like microbiota may ameliorate both gut and behavioral disturbances. FMT offers a direct method of replacing an aberrant gut microbial community with a healthier one [116]. It has already demonstrated robust efficacy in treating refractory *Clostridioides difficile* infection by reestablishing microbial diversity function. Moreover, its therapeutic effects in other conditions such as ulcerative colitis, further support the notion that the gut microbiome can have a causal role in disease [51]. Applying similar logic to ASD, FMT may represent a promising intervention for targeting core symptoms and comorbidities, though further research is required to confirm its safety, efficacy, and long-term effects in this population.

A pioneering open-label study by Kang et al. [8] implemented a modified FMT protocol (Microbiota Transfer Therapy, MTT) in 18 children with ASD. The protocol included a 2-week vancomycin course, bowel cleanse, and high-dose FMT for 7 to 8 weeks, followed by lower maintenance doses [8].

The results of early investigations on FMT in ASD have been notable. GI symptoms such as constipation and diarrhea improved by approximately 80%, with many of these improvements sustained even two years after treatment. In addition to GI relief, moderate improvements in ASD-related behaviors (such as social responsiveness and communication) were reported, based on parental assessments, and these behavioral benefits also appear to persist overtime [117]. Microbiome analyses post-MTT, revealed increased microbial diversity and a rise in beneficial bacterial populations including *Bifidobacterium*, *Prevotella* and *Desulfovibrio*. Importantly, FMT was well tolerated, with only mild and transient side effects, such as low-grade fever or gastrointestinal discomfort, and no serious adverse events were observed [8]. These results suggest that FMT may represent a promising approach for addressing both GI and behavioral problems in individuals with ASD, but further controlled research is needed to ensure efficacy and safety. Another study showed that FMT increased Akkermansia and other beneficial strains while reducing ASD severity scores over the course of 8 weeks [117].

FMT is generally safe, although there are certain concerns. Its known adverse effects include GI upset, fever, or rare transmission of infections if donor screening fails. FMT is typically reserved for severe dysbiosis when other interventions fail, given its invasive nature (usually delivered via colonoscopy, enema, or oral capsules) [118].

### 10.5. Antibiotics

The use of antibiotics in ASD has a dual perspective. Historically, some antibiotics have improved ASD symptoms (suggesting microbial involvement), yet antibiotics can also cause dysbiosis and thus potentially worsen MGB axis function [119].

The most well-known antibiotic trial in ASD involved oral vancomycin [120]. Vancomycin is thought to decrease *Clostridiales*, which potentially produce neurotoxins or propionic acid. Indeed, vancomycin use has been associated with increased *Akkermansia* (beneficial mucus degrader) and reduced ASD behaviors for the treatment duration. But long-term antibiotic use is not viable due to potential resistance and disruption of the microbiome [121].

Conversely, early-life antibiotic exposure is a risk factor for microbiome disturbance and has been linked to later-life behavioral changes in animal studies. A Finish birth cohort found that antibiotic exposure in the first year was associated with slightly higher odds of neurodevelopmental issues, though confounders exist [122].

A widely cited single case report by Rodakis described an interesting observation in which a father noticed a marked improvement in his son’s autism symptoms during a course of antibiotic treatment for an ear infection. The child exhibited better communication, fewer repetitive behaviors, and improved eye contact; however, these changes regressed after the antibiotic course ended. Although anecdotal and based on a single, non-clinical observation, this case sparked significant interest in the potential link between the gut microbiome and ASD. It suggested the possibility that altering microbial composition, intentionally or unintentionally, could influence neurobehavioral outcomes, thereby encouraging further research into microbiota-targeted therapies in ASD [123].

Antibiotics are accompanied by adverse effects. They typically eliminate beneficial bacteria and may facilitate the development of opportunistic infections, such as yeast overgrowth. Kantarcioglu et al. [124] found increased yeast levels in children with ASD, which may be connected to prior antibiotic exposure. Excessive antibiotic use may exacerbate GI disorders or induce new conditions, such as *Clostridium difficile* colitis [124].

Antibiotics are not a standard treatment for ASD, but in cases of documented microbial infections or overgrowth (e.g., culture-confirmed *Clostridia* infection), carefully administered antibiotics might be used as part of a broader strategy, ideally followed by probiotics or FMT to rebuild flora [124].

### 10.6. Nutrition and Diet

Diet is a major determinant of gut microbiota composition, and many children with ASD have atypical diets, often high in processed carbs, low in fruits and vegetables, food selectivity, and sensory difficulties [125]. Nutritional interventions thus aim to correct deficiencies, remove potential offenders, and preserve a favorable microbiome.

### 10.7. Mediterranean Diet

The Mediterranean diet emphasizes a high intake of fiber-rich foods, such as whole grains, legumes, fruits and vegetables, along with healthy fats like olive oil and lean proteins, particularly from fish. This dietary pattern supports eubiosis by promoting the growth of SCFA-producing bacteria such as *Prevotella* [126]. Encouraging a Mediterranean-style diet in individuals with ASD may help increase microbial diversity, reduce gut inflammation, and enhance the production of anti-inflammatory metabolites, potentially supporting both gastrointestinal and neurobehavioral health [127].

### 10.8. Ketogenic Diet

The ketogenic diet, which is high in fat and extremely low in carbohydrates, is often used in cases of refractory epilepsy, a condition that co-occurs in some children with ASD [50]. This diet has been shown to alter the gut microbiome, typically increasing species such as *Akkermansia* and *Desulfovibrio*. A recent report suggested behavioral, cognitive and emotional improvements in some children with ASD on a ketogenic diet, possibly due to the neuroactive properties of ketones or changes in microbial composition [128]. However, this diet can be challenging to maintain and is associated with potential side effects, including constipation and nutrient imbalances.

### 10.9. Gluten Free, Casein Free (GFCF) Diet

The GFCF diet involves the removal of gluten from wheat and other grains and casein from dairy [129]. It is a widely adopted intervention in ASD based on the theory that some individuals may have peptide intolerances, adverse physiological or behavioral responses to incompletely digested food-derived peptides, such as gluten and casein or increased intestinal permeability, allowing opioid-like peptides such as casomorphins to affect behavior [130]. Eliminating these proteins may also alter the gut microbiome by reducing populations such as *Clostridia*, which thrive on protein substrates [131]. While some studies report improvements in social behavior and language, others show no effect, and, overall, the evidence remains mixed [130]. Additionally, eliminating dairy can reduce calcium intake, raising concerns about bone health if not properly supplemented.

### 10.10. High Fiber Diet

Increasing dietary fiber by consuming whole grains, fruits, and vegetables can promote the growth of beneficial gut microbes and reduce inflammation. High-fiber diets support the production of SCFAs and enhance gut barrier function. In contrast, low-fiber and high-sugar diets may foster dysbiosis, including Candida overgrowth and bile-tolerant bacteria [132]. Many children with ASD tend to favor starchy or sugary foods while rejecting vegetables, making it difficult to implement a fiber-rich diet. In such cases, targeted fiber supplementation may serve as an effective alternative [133].

### 10.11. Omega-3 Fatty Acids

Omega-3 fatty acids, particularly EPA and DHA, have well documented anti-inflammatory effects and may benefit both the gut microbiome and brain function [134]. One pilot study in Singapore found that omega-3 supplementation improved social skills in some children with ASD [135]. While more research is needed, omega-3s are considered relatively safe and potentially beneficial adjuncts in ASD nutritional management.

### 10.12. Specific Carbohydrate Diet (SCD)/Low FODMAP Diet

These diets restrict certain types of fermentable carbohydrates to reduce GI symptoms and microbial imbalances. The SCD eliminates complex carbohydrates believed to feed harmful bacteria, while the low FODMAP diet targets specific fermentable sugars that can cause bloating and discomfort [136]. A randomized pilot trial found that while a low FODMAP diet improved GI symptoms, it did not change behavior compared to controls [137]. Although some families of children with ASD report anecdotal improvements in stool consistency and GI symptoms with these diets, robust clinical evidence supporting their efficacy is lacking. However, given the restrictive nature of these diets, they should be used carefully and ideally under professional supervision to avoid nutritional deficiencies [138].

### 10.13. Precision Diets via the Microbiome

An emerging frontier in ASD management involves the development of precision diets tailored to individual gut microbiome profiles [139]. Through stool analysis, clinicians and researchers can identify specific microbial imbalances or deficiencies. For instance, if a child with ASD has low levels of butyrate-producing bacteria, target prebiotic supplementation can be used to support their growth. Conversely, if there are signs of yeast overgrowth, such as elevated *Candida* species, dietary strategies may include reducing sugar intake and possibly incorporating antifungal agents alongside probiotics. This personalized, microbiome-informed approach holds promise for optimizing gut health and potentially influencing behavioral and cognitive outcomes in ASD, though it remains an area of ongoing research [80].

### 10.14. Postbiotics

Postbiotics are preparations composed of inactivated microorganisms or their components. They are recognized for their stability, safety, and ability to reinforce intestinal epithelial integrity, modulate immune responses and inhibit pathogenic microbes [140]. Postbiotics are more stable than probiotics because they consist of inactivated microbial cells or their components, they do not require viability to be effective, making them more resistant to storage conditions and fluctuations in temperature or pH [141]. In preclinical models of ASD, the combination of a prebiotic (α-lactalbumin) and a postbiotic (sodium butyrate) significantly improved sociability and memory compared to each alone, suggesting that targeting the MGB axis through postbiotics may offer neuroprotective potential in ASD [142].

Based on the studies reviewed in this narrative synthesis, Table 2 summarizes the main microbiota-targeted interventions in ASD, the main findings, study types, sample size ranges and an overall qualitative grade of evidence strength.

As shown in Table 2, probiotics and dietary intervention currently have the largest evidence base, though heterogeneity in study protocols limits firm conclusions. Prebiotics, symbiotics and FMT/MMT remain promising, but under exploration, while antibiotic-based approaches show only transient benefits. Larger, well-controlled studies are essential to confirm efficacy and assess long-term safety.

Table 3 further summarizes the main dietary approaches investigated in relation to the MGB axis and ASD. Based on the studies reviewed in this narrative synthesis, for each diet, the proposed mechanisms, reported benefits and potential concerns are outlined, offering a comparative overview of their evidence base and clinical considerations.

As shown in Table 3, gluten-free/casein-free and ketogenic diets are the most widely studied, though evidence remains heterogeneous. Mediterranean and high-fiber dietary patterns appear to promote microbial diversity and exert anti-inflammatory effects, while omega-3 fatty acid supplementation has been linked to potential improvements in behavioral outcomes. Nevertheless, most dietary interventions are limited by small sample sizes, variability in protocols and inconsistent findings. Larger, well-controlled studies are required to confirm their efficacy, assess their long-term safety, and evaluate their applicability in clinical practice.

## 11. Conclusions and Future Perspectives

The exploration of the MGB axis in ASD has illuminated a pivotal intersection of neuroscience, gastroenterology, and microbiology. The evidence compiled in this review underscores that the gut microbiome, far from being an isolated entity, significantly interacts with neurodevelopmental processes. Alterations in gut bacteria, whether in composition, diversity or metabolic function, are increasingly linked to the complex symptomatology of ASD, from core behavioral features to the high prevalence of gastrointestinal comorbidities. While animal studies may suggest possible causal mechanisms, most human evidence to date remains correlational.

The fact that ASD-related microbiota findings vary considerably between studies underscores the influence of multiple confounding factors, including geographic variation, dietary patterns, methodological differences, age, medication exposure and early-life influences such as mode of delivery and maternal health. Recognizing these sources of variability is essential for interpreting the existing results and for designing standardized and reproducible studies in the future.

To date, research indicates that gut dysbiosis in individuals with ASD can trigger a cascade of physiological disturbances. Increased intestinal permeability, commonly known as “leaky gut”, allows the entrance of microbial metabolites and endotoxins into the bloodstream, which causes systemic and neuroinflammatory reactions. These processes in turn may exacerbate or contribute to the neurological alterations frequently met in ASD. The presence of FGIDs in individuals with ASD is not just a coincidence; instead, it suggests a disruption of gut–brain homeostasis. In short, many individuals with ASD appear to present a cycle of gut–brain dysregulation in which stress exacerbates GI disorders, and, in turn, these disorders amplify stress-related responses, with the MGB axis acting as a mediator.

The MGB axis represents a viable area for therapeutic intervention. Dietary modification and nutritional support are low-risk strategies to modulate microbiome, with potential benefits for both GI health and behavior. Interventions including probiotics, prebiotics, and symbiotics have shown promise in improving GI disturbances and certain behavioral issues, although the results may vary across individuals and studies. More experimental approaches, including FMT and MTT, have shown substantial benefits in GI health and behavioral improvements in early trials. These strategies need further evaluation in well-controlled studies. While antibiotic use is not a standard treatment for ASD, evidence suggests that targeting particular gut bacteria may temporarily alleviate symptoms. This supports the hypothesis that certain microbial metabolites may play a role in ASD-related behaviors, although current evidence remains preliminary and requires further validation.

Despite these promising avenues, it is essential to approach the MGB axis ASD link with scientific caution. The heterogeneity in ASD and its microbiota findings implies that one size may not fit all. What helps one subset of individuals might be ineffective or even harmful in another. Thus, future research should prioritize identifying biomarkers (microbial, metabolic, or genetic) that predict which individuals are most likely to benefit from microbiome-targeted therapies. Long-term and large-scale studies are needed to establish causal relationships such as determining if dysbiosis causes certain ASD symptoms or if it is a side effect and to evaluate the sustained efficacy and safety of interventions like probiotics or FMT beyond short follow-up periods. In addition, a key methodological limitation of this review is the absence of standardized critical appraisal or study quality, which restricts the ability to formally weigh the strength of evidence presented.

Another intriguing future direction is investigating how the MGB axis might interact with other factors in ASD, such as genetics or environmental exposure. For instance, could certain gene variants in ASD affect how the body responds to dysbiosis? Additionally, the potential for early intervention is worth exploring: can modifying the microbiota in infancy Via diet, probiotics, or even maternal interventions during pregnancy reduce the risk of severe ASD in genetically predisposed children?

In conclusion, the MGB axis has moved from the periphery to the center of ASD research, offering not only a deeper understanding of ASD pathophysiology but also tangible therapeutic targets. Ensuring a healthy “second brain” in the gut could become a key component of comprehensive ASD management. Realizing this vision will require continued interdisciplinary collaboration, carefully designed clinical trials, and a personalized approach to therapy. The hope is that by harnessing the gut–brain connection, we can open new pathways to improve the lives of individuals with ASD and their families.

## Figures and Tables

**Table 1 nutrients-17-02984-t001:** Selected key studies reporting gut microbiota alterations in ASD.

Study (Year)	Population/Sample	Key Findings on Gut Microbiota in ASD
De Angelis et al. (2013, cross sectional) [46]	Children with ASD vs. TD controls	↑ *Clostridium*, *Desulfovibrio*, and *Barnesiella intestinihominis*; ↓ *Bifidobacterium* and *Akkermansia*; altered fecal metabolomic profile.
Kang et al. (2013, cross sectional) [8]	Children with ASD vs. TD controls	↓ *Prevotella*, *Coprococcus*, and overall microbial diversity; changes associated with gastrointestinal symptom severity.
Tomova et al. (2015, cross sectional) [47]	Children with ASD vs. TD controls	↑ *Lactobacillus* spp. and *Desulfovibrio*; ↓ *Bifidobacterium*; significant differences in metabolic profiles.
Son et al. (2015, cross sectional) [48]	Children with ASD vs. sibling controls	↑ *Sutterella* and *Ruminococcus* in ASD; distinct gut microbial community structure observed.
Sharon et al. (2019, preclinical animal study) [32]	FMT from ASD donors to germ-free mice	Mice receiving ASD donor microbiota displayed autism-like behaviors; ↑ *Clostridiaceae*, *Bacteroides* and *Lactobacillales*, and ↓ microbial diversity.
Iglesias-Vázquez et al. (2020) [49]	Meta-analysis of 18 studies (children with ASD vs. controls)	↓ *Bifidobacterium* and *Firmicutes*; ↑ *Lactobacillus* and some *Clostridia* in ASD; significant heterogeneity noted.
Sivamaruthi et al. (2020) [50]	Narrative review	Common features in ASD: ↑ *Akkermansia*, *Clostridium*, *Desulfovibrio*, *Sutterella* and *Faecalibacterium*; ↓ *Bifidobacterium*, *Prevotella*, and overall *Firmicutes*.

Note: ASD = Autism Spectrum Disorder, TD = Typically Developing, ↑ = increased, ↓ = decreased, FMT = fecal microbiota transplantation.

**Table 2 nutrients-17-02984-t002:** Summary of evidence for microbiome-targeted interventions in ASD.

Intervention	Main Findings in ASD	Study Design(s)	Sample Size Range
Probiotics	Improved GI symptoms; some studies also report behavioral improvements; strain- and dose-specific effects.	RCTs, open-label trials, meta-analyses	20–141
Prebiotics	Improved GI function; limited behavioral effects; effects may depend on baseline microbiota composition.	RCTs, pilot studies	17–71
Symbiotics	Potential additive effect on GI health when combined with probiotics; minimal behavioral impact reported.	Small RCTs	25–48
FMT/MTT	Improved GI and some behavioral symptoms; sustained effects in follow-up; safety profile generally acceptable.	Open-label trials, pilot studies	18–28
Antibiotics	Temporary reduction in GI and behavioral symptoms; effects not sustained; potential adverse effects.	Small open-label trials	11–13
Dietary Interventions	Gluten-free/casein-free diets may improve GI and behavioral symptoms in some individuals; results inconsistent.	RCTs, observational studies, meta-analyses	15–102

Note: ASD = Autism Spectrum Disorder, GI = gastrointestinal, RCT = Randomized Controlled Trial, FMT = fecal microbiota transplantation, MTT = Microbiota Transfer Therapy.

**Table 3 nutrients-17-02984-t003:** Summary of dietary approaches investigated in relation to the MGB axis and ASD.

Dietary Approach	Proposed Mechanisms	Reported Benefits	Potential Concerns
Gluten Free/Casein Free Diet (GFCF)	Reduces intake of gluten and casein thought to exacerbate GI and behavioral symptoms Via opioid-like peptides.	Some studies report improvements in behavior and GI symptoms	Evidence inconsistent; risk of nutritional deficiencies if not supervised
Ketogenic Diet	Shifts metabolism toward ketone body production; proposed effects on neurotransmission and mitochondrial function	Reported improvements in seizure control and some behavioral domains.	Restrictive adherence challenging, potential nutrient deficiencies
Specific Carbohydrate Diet (SCD)	Eliminates complex carbohydrates to limit substrates for pathogenic gut microbes	Anecdotal improvements in GI function reported by families	Limited formal evidence; highly restrictive; risk of imbalance
Low FODMAP Diet	Reduces fermentable oligosaccharides, disaccharides, monosaccharides and polyols to improve GI symptoms	Can reduce bloating, abdominal pain and stool irregularities	Evidence in ASD limited; requires professional supervision to avoid deficiencies
Mediterranean Diet	Rich in fiber, polyphenols, and healthy fats; promotes eubiosis and anti-inflammatory effects	Associated with improved gut microbiota diversity and systemic health	Few ASD-specific trials; adherence varies across cultures
Omega-3 Fatty Acids	Anti-inflammatory effects; modulate neuronal membrane fluidity and signaling	Some evidence of improved hyperactivity and social functioning	Mixed trial results; effect sizes small; supplementation variability
High-Fiber Diets	Enhance SCFA production; improve gut barrier function; support microbial diversity	May improve bowel habits and support favorable microbiota profiles	Few ASD-specific studies; GI tolerance varies

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
