# Peer review of "Microbiota Gut–Brain Axis and Autism Spectrum Disorder: Mechanisms and Therapeutic Perspectives"

_nutrients, 2025, doi:10.3390/nu17182984_

Round 1
Reviewer 1 Report
Comments and Suggestions for Authors
The manuscript provides a review on the role of the microbiota–gut–brain (MGB) axis in Autism Spectrum Disorder (ASD). The text addresses mechanistic pathways, gut microbiota alterations, gastrointestinal comorbidities, and finally therapeutic approaches. It is well-structured and organized, and contains a rich reference base covering human, animal, and mechanistic studies. No overstatement of therapies' benefits is observable. The topic is of high interest given the growing clinical and research focus on microbiome-based interventions in neurological disorders.
However, while the authors summarize a wide range of literature, their conclusions mainly describe findings rather than critically evaluating the quality of the evidence. At times, they suggest a causal relationship between dysbiosis (microbiota imbalance) and ASD without sufficiently backing it up. Improvements could be made in the language to clarify these points.
Major comments
While the authors describe their literature search strategy, there is no formal quality assessment of included studies, which limits the ability to weigh the strength of evidence. Please, include a critical appraisal of study quality or state the absence of such evaluation as a methodological limitation in the discussion.
In several places, the text suggests or strongly implies a causal role of dysbiosis in ASD (e.g., “dysbiosis… probably makes things worse by causing leaky gut and an overactive immune system” [lines 350–355]). While animal models provide some causal evidence, human studies are still mostly correlational. Use more cautious language (e.g., “is associated with,” “may contribute to”) and clearly distinguish between correlation and causation.
The authors note inconsistencies (e.g., Bacteroidetes/Firmicutes ratios) but do not explore possible sources in depth (diet, geography, sequencing methods, age, medication). Discussion on why results differ across studies could be expanded.
Although multiple interventions are covered, the evidence base for many (e.g., probiotics, ketogenic diet, GFCF diet) is small and heterogeneous, with inconsistent outcomes. A brief summary table grading the level of evidence for each intervention (e.g., strong/moderate/weak), based on study design and sample size, should be added.
Section 10 omits postbiotics, which are increasingly recognized for their stability, safety, and potential neuroprotective effects via the MGB axis. Including a brief note on their mechanisms and emerging evidence in ASD would make the therapeutic overview more complete.
Minor comments
Several grammatical errors and repetitions are found across the text (e.g., “newly new activation,” “problem with their in their gut barrier,” repeated explanations of SCFA effects). A thorough language edit would improve precision.
Some sections (e.g., dietary interventions, precision diets) would improve by citing studies from 2023–2025.
Table 1 is informative, but could include an extra column for study type (cross-sectional, RCT, meta-analysis) to visualize evidence strength quickly.
Use consistent formatting for bacterial names (italicized genus/species) and abbreviations (define at first use).
Occasional typographical errors:
Previously hyphenated words appearing split in the text:
- Lines 390 and 417: “Clos-tridiaceae” and “Clos-tridium” should be “Clostridiaceae" and “Clostridium”
- Line 505: "hip-pocampal" should be "hippocampal."
- Line 605: "Ak-kermansia" should be "Akkermansia."
Other misspellings
- Lines 133, 135 and 265: "SFCAs” should be “SCFAs” (short-chain fatty acids).
- Line 364: "editable bowel syndrome" should be “irritable bowel syndrome.”
- Line 369: “FFIDs” should be “FGIDs” (Functional Gastrointestinal Disorders).
- Line 463: " such Lactobacillus and Bifidobacterium species" should be "such as Lactobacillus and Bifidobacterium species."
- Line 512: "triads” should be “trials.”
- Line 518: “germs” is a colloquial term; “microorganisms” or “bacteria” would be more appropriate.
- Line 623: “such yeast overgrowth” should be “such as yeast overgrowth.”
- Line 629: “Clostridia infection”; do you mean “Clostridium infection.”
- “can it be used to support ..... ” should be “it can be used to support ....”
- Line 653: “Glute Free” should be “Gluten Free.”
The manuscript is generally readable, with a clear structure, but it contains recurrent grammatical errors, strange phrasing, and stylistic inconsistencies.
Examples:
- Redundant or imprecise expressions: "newly new activation", "diet causing gluten free"
- Occasional syntax issues and misplaced modifiers.
- Shifts in tone between formal scientific language and conversational phrasing (e.g., “This means that something about the ASD microbial community can cause features…”).
- Colloquial expressions. "probiotics are living microorganisms that are good for you."
A professional language edit or proofreading pass is recommended to ensure consistent clarity and precision throughout the manuscript.
Author Response
Response to Reviewer 1 Comments
We appreciate very much the Reviewers’ interest in our work, for the helpful comments as well as the effort and time put into the review of this manuscript. All comments have been carefully considered and responded point by point.
Major Comment 1: While the authors describe their literature search strategy, there is no formal quality assessment of included studies, which limits the ability to weigh the strength of evidence. Please, include a critical appraisal of study quality or state the absence of such evaluation as a methodological limitation in the discussion.
Response 1: We thank the Reviewer for this comment and we appreciate the observation. Given the narrative review design, a formal quality assessment was not conducted. In the revised manuscript, we have explicitly acknowledged this a methodological limitation in the Discussion section and have clarified that the weighting of evidence was based on study design, sample size, and recurrence of findings, but without the use of standardized appraisal tool.
Location in Revised Manuscript:
Section 2 - Review Protocol, last paragraph “ Given the narrative nature of this review, no formal quality appraisal of induced studies was conducted. However, started with high levels of evidence (e.g., meta-analyses, controlled trials) where prioritized, and consensus or recurring findings were emphasized. Seminal older studies were also included to provide historical context where appropriate. (lines 101-105)
Section 11 - Conclusion and Future Perspective, we added one sentence noting this limitation “A key methodological limitation of this review is the absence of standardized critical appraisal of study quality, which restricts the ability to formally weight the strength of evidence presented.” (lines 793-795)
Major comment 2: In several places, the text suggests a causal role of dysbiosis in ASD without sufficient evidence. Use more cautious language and distinguish between correlation and causation.
Response 2: We thank the Reviewer for this comment. We have carefully revised the manuscript to replace causal language with more cautious terms such as “is associated with”, “may contribute to”, or “might influence”, particularly in sections describe human studies. We have explicitly distinguished between findings from animal models (which may support causality) and human studies (largely correlational).
Location in Revised Manuscript:
Section 6 – Functional Implications of Dysbiosis in ASD. We revised the sentence “This suggests that certain characteristics of the ASD microbial community may contribute to features that are similar to those of ASD, although the specific microbes and mechanisms are still being investigated [58].” (lines 283-285).
Section 7.1 – Dysbiosis and Intestinal Barrier Dysfunction. We revised the sentence “effect of dysbiosis is compromised gut epithelial barrier (lines 295-296).
Section 7.3 – Metabolic Endotoxemia and Brain Impact. We revised the sentence “These findings highlights how microbial metabolites, typically confined within the gut-liver system, may enter the bloodstream and potentially influence modulate neurodevelopment and behavioral outcomes (lines 347-350).
Section 8 – Leaky gut and BBB. We revised the sentence “Dysbiosis in ASD is not a benign bystander, though it likely contributes to pathophysiological processes through leaky gut and heightened immune activation” (lines 364-365).
Section 9 – Microbiota-FGID Link. We revised the sentence “leading to diarrhea or pain gastrointestinal discomfort” (lines 428).
Section 11 – Conclusion and Future Perspective. We revised the sentence “Alterations in gut bacteria, whether in composition, diversity or metabolic function, are increasingly linked to the complex symptomatology of ASD, (lines 753-755), and we added the sentence “ While animal studies may suggest possible causal mechanisms, most human evidence to date remains correlational (lines 756-757).
Major Comment 3: Discussion on inconsistencies (e.g., Bacteroidetes/Firmicutes ratios) could be expanded to address possible sources (diet, geography, sequencing methods, age, medication).
Response 3: We thank the Reviewer for this comment. We have incorporated the requested paragraph discussing the possible sources of inconsistencies in microbiota alteration in Section 5. The revised text now addresses geographic variation, dietary patterns, methodological differences, age-related factors, medication exposure, and early-life influences such as mode of delivery and maternal health, with appropriate references. The revised paragraph “ Inconsistencies in reported microbiota alterations, such as differences in the Bacteroidetes/Firmicutes ration, may stem from multiple factors. Geographic variation can influence microbial profiles through differences in environmental exposures and cultural dietary practices [52]. Dietary patterns, including macronutrient composition and food processing, are known to shape the gut microbiota and may differ substantially between study populations [53]. Methodological factors, such as DNA extraction protocols, sequencing platforms and bioinformatic pipelines can also affect taxonomic results [54]. In addition, microbiota composition changes naturally with age, and many studies include participants with a wide age range, which may confound results [55]. Medications exposure, particularly antibiotics, probiotics and psychotropic drugs, can alter microbial communities and is not consistently controlled [56]. In addition, perinatal and maternal influences such as mode of delivery, feeding type and maternal health conditions (e.g. stress, diet, gestational diabetes), play a role in early microbiota development [29]. can be found between lines 258-271.
Major Comment 4: Although multiple interventions are covered, the evidence base for many (e.g., probiotics, ketogenic diet, GFCF diet) is small and heterogeneous, with inconsistent outcomes. A brief summary table grading the level of evidence for each intervention (e.g., strong/moderate/weak), based on study design and sample size, should be added.
Response 4: We thank the Reviewer for this comment. We have added a new table (Table 2) in section 10 summarizing each intervention (probiotics, prebiotics, symbiotics, FMT, antibiotics, dietary intervention), its main findings in ASD, study type, sample size range and a qualitive grade of evidence strnght (strong/moderate, weak) based on study design and sample size (line 720).
In addition, following the table, we included a concluding sentence to synthesize the key insights: "As shown in Table 2, probiotics and dietary interventions currently have the largest evidence base, though heterogeneity in study protocols limits firm conclusions. Prebiotics, symbiotics, and FMT/MTT remain promising but underexplored, while antibiotic-based approaches show only transient benefits. Larger, well-controlled studies are essential to confirm efficacy and assess long-term safety." (lines 725-729)
|
Major Comment 5: Section 10 omits postbiotics, which are increasingly recognized for their stability, safety, and potential neuroprotective effects via the MGB axis. Including a brief note on their mechanisms and emerging evidence in ASD would make the therapeutic overview more complete. Response 5: We thank the Reviewer for this comment. We have added a new subsection (10.14) on postbiotics summarizing the definition, mechanism of action, advantages over probiotics and current possible evidence over ASD (preclinical model stydies). The revised paragraph “Postbiotics are preparations compromised of inactivated microorganisms or their components and they are recognized for their stability, safety and ability to reinforce intestinal epithelial integrity, modulate immune responses and inhibit pathogenic microbes [137]. Postbiotics are more stable than probiotics because they consist of inactivated microbial cells or their component, they do not require viability to be effective, making them more resistant to storage conditions and flucutations in temperature or pH [138]. In preclinical models of ASD, the combination of a prebiotic (α-lactalbumin) and a postbiotic (sodium butyrate) significantly improved sociability and memory compared to each alone, suggesting that targeting the MGB axis through postbiotics may offer neuroprotective potential in ASD [139].” can be found between lines 706-715.
Minor comment 6: Several grammatical errors and repetitions are found across the text (e.g., “newly new activation,” “problem with their in their gut barrier,” repeated explanations of SCFA effects). A thorough language edit would improve precision.
Response 6: We thank the Reviewer for this comment. All such errors have been corrected, and redundant phrases mainly in the SCFA effects in Section 10 have been removed for conciseness and clarity.
Minor comment 7: Some sections (e.g., dietary interventions) would improve by citing studies from 2023–2025.
Response 7: We thank the Reviewer for this comment. We have updated the literature with recent studies (2022–2025) in sections on dietary interventions.
Minor Comment 8: Table 1 is informative, but could include an extra column for study type (cross-sectional, RCT, meta-analysis) to visualize evidence strength quickly. Response 8: We thank the Reviewer for this comment. We have updated Table 1 by appending the study design type next to each referenced work directly (e.g., cross-sectional, meta-analysis, animal model, narrative review), providing immediate clarity regarding the level of evidence. This approach avoids adding a new table column while enhancing transparency and helping readers quickly assess the evidence hierarchy.
Minor Comment 9: Use consistent formatting for bacterial names (italicized genus/species) and abbreviations (define at first use). Response 9: We thank the Reviewer for this comment. All bacterial genus and species names are now italicized, and abbreviations are defined at first mention and used consistently thereafter.
Minor Comment 10: Typographical errors: “Clos-tridiaceae” → “Clostridiaceae,” “hip-pocampal” → “hippocampal,” “Ak-kermansia” → “Akkermansia,” “SFCAs” → “SCFAs,” “editable bowel syndrome” → “irritable bowel syndrome,” “FFIDs” → “FFIDs,” “such Lactobacillus” → “such as Lactobacillus,” “triads” → “trials,” “germs” → “microorganisms,” “such yeast overgrowth” → “such as yeast overgrowth,” “Clostridia infection” → “Clostridium infection,” “can it be used…” → “it can be used…,” “Glute Free” → “Gluten Free.”
|
Reviewer 2 Report
Comments and Suggestions for Authors
This a very interesting review regarding microbiota gut-brain axis in autism, however adjustments are needed before publication.
Here my suggestions:
-
Line 17 – "major scientific databases" should be named explicitly here for transparency.
-
Line 22 – The phrase "provided draft document" is unclear. Please clarify or rephrase.
-
Line 29 – Typo: "newly new activation" is grammatically incorrect; please revise.
-
Line 38 – "gluten causing free" appears to be a typo; likely meant "gluten-free".
-
Line 44 – Add a comma after "However" for correct punctuation.
-
Line 62 – Consider replacing "reverberate" with a more formal scientific term such as "affect" or "influence".
-
Line 94 – "Both human studies clinical and epidemiological…" needs restructuring for grammar.
-
Line 98 – Typo: "composition differenced" maybe it should be "composition differences".
-
Line 138 – Consider citing more recent studies regarding the kynurenine pathway.
-
Line 149 – Avoid the word "problems"; use "dysfunctions" or "disorders" for precision.
-
Line 187 – Typo: "Homestasis" it should be "Homeostasis".
-
Line 194 – Clarify what is meant by “availability of their precursors”...please explain better.
-
Line 259 – I suggest to add a sentence to reinforce whether changes are causal or consequential to ASD.
-
Line 291 – "real problem with their in their gut barrier" contains redundancy; please correct.
-
Line 330 – Ensure proper referencing of behavioral studies on propionic acid, and deep better the topic
-
Line 364 – "editable bowel syndrome" ??? is likely a typo; maybe "irritable bowel syndrome".
-
Line 430 – Citation format for "Janina, R.; et al., 2021" inconsistent with others—revise accordingly.
-
Line 498 – Suggest adding a disclaimer about the current low evidence level supporting probiotic use in ASD.
-
Line 543 – Typo: "meals low in MACs", it is better "diets low in MACs".
-
Line 560 – Define "MD" at first mention (likely meant to be "FMT").
-
Line 589 – Please clarify donor screening protocols to address safety concerns of FMT.
-
Line 599 – Citation, the number 109, is needed for Sandler et al.’s vancomycin trial also in the first sentence.
-
Line 613 – Rodakis case is anecdotal; recommend clarifying its non-clinical nature.
-
Line 655 – Clarify what is meant by "peptide intolerances"; provide a supporting citation.
-
Line 685 – Note the lack of strong evidence for SCD/Low-FODMAP diets in ASD.
-
Line 723 – I recommend toning down certainty around microbial metabolites playing a role in ASD behaviors.
Moderate editing is required
Author Response
Response to Reviewer 2 Comments
We appreciate very much the Reviewers’ interest in our work, for the helpful comments as well as the effort and time put into the review of this manuscript. All comments have been carefully considered and responded point by point.
Comment 1: "major scientific databases" should be named explicitly here for transparency.
Response 1: We thank the Reviewer for this comment. Literature searches were conducted in the Pubmed, Scopus and Google Scholar databases. We have revised the manuscript “searching major scientific databases including PubMed, Scopus, and Google Scholar” (lines 17-18)
Comment 2: The phrase "provided draft document" is unclear. Please clarify or rephrase.
Response 2: We thank the Reviewer for this comment. We have clarified this statement as follows: “Original references from the initial manuscript draft were retained and supplemented for comprehensiveness and accuracy.” (lines 23-24).
Comment 3: Typo: "newly new activation" is grammatically incorrect; please revise.
Response 3: We thank the Reviewer for this comment. We have added the revised sentence “newly activated”. (line 30)
Comment 4: "gluten causing free" appears to be a typo; likely meant "gluten-free".
Response 4: We thank the Reviewer for this comment. We have added the revised sentence “high fiber, gluten free, or ketogenic diet” (line 39).
Comment 5: Add a comma after "However" for correct punctuation.
Response 5: We thank the Reviewer for this comment. We have added the comma after the word “However” (line 45)
Comment 6: Consider replacing "reverberate" with a more formal scientific term such as "affect" or "influence".
Response 6: We thank the Reviewer for this comment. We have replaced the word reverberate and the revised sentence is “can affect the MGB axis, affecting brain development, behavior, and cognition” (line 64)
Comment 7: "Both human studies clinical and epidemiological…" needs restructuring for grammar.
Response 7: We thank the Reviewer for this comment. We have revised the sentence “Both clinical and epidemiological human studies, as well as mechanistic animal and in vitro studies were considered”. (line 94-95)
Comment 8: Typo: "composition differenced" maybe it should be "composition differences.
Response 8: We thank the Reviewer for this comment. We have revised the sentence “microbiota composition differences in ASD” (line 98).
Comment 9: Line 138 – Consider citing more recent studies regarding the kynurenine pathway.
Response 9: We thank the Reviewer for this comment. We have updated with more recent study considering the kynurenine pathway. The revised sentence “Tryptophan metabolites from gut bacteria or the host's kynurenine pathway produce neuroactive compounds that alter serotonergic transmission, while gut-derived tryptophan is involved in processes that change neural circuits and influence brain function and behavior” can be found between lines 139-142.
Comment 10: Line 149 – Avoid the word "problems"; use "dysfunctions" or "disorders" for precision.
Response 10: We thank the Reviewer for this comment. We have revised the wording by replacing “problems” with other suitable words, providing a more accurate and discipline-appropriate description.
Comment 11: Line 187 – Typo: "Homestasis" it should be "Homeostasis".
Response 11: We thank the Reviewer for this comment. We have revised the word "Homestasis" to "Homeostasis".
Comment 12: Clarify what is meant by “availability of their precursors”...please explain better.
Response 12: We thank the Reviewer for this comment. The phrase “availability of their precursors” refers to the presence and sufficient supply of dietary and endogenous compounds required for the synthesis of the respective neurotransmitters. The revised sentence, in order to be more clear is “The gut microbiome also modulates central neurotransmitters levels by changing the availability of their precursors, that is, the dietary and endogenous molecules required for neurotransmitter biosynthesis” (lines 194-196).
Comment 13: Line 259 – I suggest to add a sentence to reinforce whether changes are causal or consequential to ASD.
Response 13: We thank the Reviewer for this comment. We have revised the sentence “It remains unclear if changes in microbiota are a cause or consequence of ASD (or behaviors connected to it, such a limited diet or anxiety that affects gut motility), underscoring the need for longitudinal and mechanistic studies to establish directionality [51]. (lines 255-257).
Comment 14: Line 291 – "real problem with their in their gut barrier" contains redundancy; please correct.
Response 14: We thank the Reviewer for this comment. We have reducted the duplicated word “their”.
Comment 15: Line 330 – Ensure proper referencing of behavioral studies on propionic acid, and deep better the topic
Response 15: We thank the Reviewer for this comment. We have revised the section on propionic acid to incorporate updated evidence and more detailed mechanistic insights. These additions enhance the depth and currency of the discussion. This is the revised paragraph “When the gut barrier integrity is compromised, gut derived metabolites sush as propionic acid (PPA), a short-chain fatty acid produced from gut bacteria fermentation, can cross into the systemic circulation and get to the CNS [48]. In controlled trials, administration of propionic acid in rodents, induces behavioral alteration such as repetitive behaviors, social isolation and altered motor activity [71]. Similarly, p-cresol, which derives from the fermentation of tyrosine by specific gut microorganisms, can inhibit dopamine beta-hydroxylase, which disrupts the balance of neurotransmitters. Some children with ASD had elevated levels of p-cresol sulfate, particularly those with FGIDs [58]. These findings highlights how microbial metabolites, typically confined within the gut-liver system, may enter the bloodstream and potentially influence modulate neurodevelopment and behavioral outcomes [72].” (lines 340-350)
Comment 16: Line 364 – "editable bowel syndrome" ??? is likely a typo; maybe "irritable bowel syndrome".
Response 16: We thank the Reviewer for this comment. This typographical error has been corrected as suggested.
Comment 17: Line 430 – Citation format for "Janina, R.; et al., 2021" inconsistent with others—revise accordingly.
Response 17: We thank the Reviewer for this comment. We have reviewed the reference “Janina, R.; et al., 2021” and were unable to locate any corresponding publication in our reference database. This appears to have been an erroneous citation. We have therefore removed it and revised the sentence to cite a valid and relevant source.
Comment 18: Suggest adding a disclaimer about the current low evidence level supporting probiotic use in ASD.
Response 17: We thank the Reviewer for this comment. We have added a disclaimer in the “Probiotics” section, clarifying that current evidence for probiotic use in ASD is limited and based on heterogeneous and small-scale studies, in line with the suggestion. The revised sentence is “However, it is important to note that the current evidence base for probiotic use in ASD remains limited, with heterogenous study designs and small sample sizes impending firm conclusion [100]. (lines 525-527)
Comment 18: Line 543 – Typo: "meals low in MACs", it is better "diets low in MACs".
Response 18: We thank the Reviewer for this comment. We have revised the word "meals" to "diet".
Comment 19: Line 560 – Define "MD" at first mention (likely meant to be "FMT").
Response 19: We thank the Reviewer for this comment. We revised the word “MD” which it meant to be “FMT”.
Comment 20 : Line 589 – Please clarify donor screening protocols to address safety concerns of FMT.
Response 20: We thank the Reviewer for this comment. We have expanded Section 10.4 with a clearer description of donor screening protocols. These updates follow the latest safety recommendations addressing emerging infectious threats. This is the revised paragraph “To address safety concerns, FMT donor screening involves a detailed medical and lifestyle history, laboratory testing of blood and stool for infectious agents and quarantine measures. Repeat testing is performed for regular donors. These steps are consistent with recent guidelines that include precautions for emerging pathogens.” (line 567-570)
Comment 21: Line 599 – Citation, the number 109, is needed for Sandler et al.’s vancomycin trial also in the first sentence.
Response 21: We thank the Reviewer for this comment. We have added the appropriate citation (Ref. 117) immediately following the first mention of Sandler et al.’s vancomycin trial to ensure clarity and consistency in referencing.
Comment 22: Line 613 – Rodakis case is anecdotal; recommend clarifying its non-clinical nature.
Response 22: We thank the Reviewer for this comment. We have revised the sentence to clarify that the Rodakis case refers to an anecdotal, non-clinical observation, and have emphasized that it should not be interpreted as clinical evidence. The revised sentence “Although anecdotal and based on a single, non-clinical observation” is located in line 621.
Comment 23: Line 655 – Clarify what is meant by "peptide intolerances"; provide a supporting citation.
Response 23: We thank the Reviewer for this comment. We have clarified the meaning of “peptide intolerances” and we have also added supporting citations. This is the revised sentence “peptide intolerances-adverse physiological or behavioral responses to incompletely digested food-derived peptides such as gluten and casein” (line 660-661).
Comment 24: Line 685 – Note the lack of strong evidence for SCD/Low-FODMAP diets in ASD.
Response 24: We thank the Reviewer for this comment. We have revised the section to emphasize that while anecdotal improvements are reported, robust clinical evidence is lacking. This is the revised paragraph “These diets restrict certain types of fermentable carbohydrates to reduce GI symptoms and microbial imbalances. The SCD eliminates complex carbohydrates believed to feed harmful bacteria, while the low FODMAP diet targets specific fermentable sugars that can cause bloating and discomfort [133]. A randomized pilot trial found that while a low FODMAP diet improved GI symptom, it did not change behavior compared to controls [134]. Although some families of children with ASD report anecdotal improvements in stool consistency and GI symptoms with these diets, robust clinical evidence supporting their efficacy is lacking. However, given the restrictive nature of these diets, they should be used carefully and ideally under professional supervision to avoid nutritional deficiencies [135].(lines 683-692).
Comment 25: Line 723 – I recommend toning down certainty around microbial metabolites playing a role in ASD behaviors.
Response 24: We thank the Reviewer for this comment. We have revised the conlusion to tone down the certainty regarding the role of microbial metabolites in ASD behaviors. This is the revised sentence “ This supports the hypothesis that certain microbial metabolites may play a role in ASD-related behaviors, although current evidence remain preliminary and requires further validation.” (lines 781-783)
Reviewer 3 Report
Comments and Suggestions for Authors
The review article presents relevant information on gut–brain interactions in the pathogenesis of ASD and discusses dietary approaches that may be suitable for individuals with ASD. However, the manuscript would benefit from the inclusion of more in-depth scientific evidence in several sections, including specific diets, their relevance to ASD, and supporting data from published studies. Improving these areas is recommended to publish a review article in a scientific journal. Additionally, a substantial portion of the text is written in a style that is more conversational than scientific, which should be revised to align with formal academic writing. Finally, the manuscript contains numerous grammatical errors that should be addressed to improve clarity and readability. Here are some of the suggestions and areas identified. Below are specific suggestions and areas identified for improvement. Major revisions are recommended, taking into account the following points.
It would be beneficial to include information about variations in gut microbiota in ASD people with a focus on inter-individual variations.
Line 269-270: provide information about the sample used for this transplantation “gut microbiota from human ASD donors”
Line 526: SCFA’s do not change how neurotransmitters are made. They may influence the levels of neurotransmitters either directly or indirectly. Rephrase the sentence for scientific accuracy.
It is suggested to include more information about different studies that have tested different types of gut microbiota-altering diets for ASD patients.
A more formal tone in writing is recommended. For example, instead of saying, “The brain releases corticotropin-releasing hormone (CRH), which tells the adrenal glands to make cortisol,” you can say, “stimulates the adrenal glands to release CRH.”
Avoid using contractions like it’s,” “don’t,” “can’t,” “won’t, isn’t etc. in scientific writing. Thoroughly review the whole review article to address these issues.
This advice applies throughout the entire review article; authors are encouraged to carefully check for informal words and replace them with more formal language.
Line 124: talking – interacts
Line 148: strange signals back to the brain: how do you define strange? You could use alterations in the signaling to the brain.
Line 191: rephrase the sentence using scientific terminology “The host, on the other hand, gives the guest food and a safe place to stay”
Line 232 – 233: Rephrase this sentence into a scientific way of writing “ In many ASD groups, the Bacteroidetes: Firmicutes ratio is out of whack“.
Lines : 384-387: Rephrase this sentence using the scientific terminology for poop. “Also, keeping poop in the body for a long time might make gut bacteria ferment more and ab- sorb more microbial toxins, which may affect behavior and brain function in people with ASD”.
Lines 407 – 408: provide proper evidence for this sentence. Can not use the word some people think when writing a scientific review “Some people think that visceral hypersensitivity and how bad FGIDs like IBS may be in people with ASD are worse since their nerve systems are already sensitive”.
Line 446: rephrase the sentence “It is at the heart of this interaction”
Line 524: nerve to help the gut and brain talk to each other. Rephrase it in a scientific way of writing as “They interact with each other”
Lines: 713-715: not a scientific way of writing “The MGB axis is a viable area for intervention from a therapeutic point of view. Changing your diet and getting nutritional assistance are low-risk ways to improve your microbiome, which might be good for both your GI health and your behavior”
Line 505: correct the word “ hip-pocampal”
Grammar:
Line 130: such lipopolysaccharide (LPS) – such as LPS
Line 132: signaling molecules are used – are often used
Line 134: such acetate, propionate, - Such as acetate, propionate
Line 134- 135: They are generated when dietary fibers are fermented.- they are generated when dietary fibers are fermented by the gut microbiota
Line 134: These SFCAs provide colonocytes energy - These SCFAs provide colonocytes with energy.
Line 138: which can come from bacteria – Produced by gut microbiota
Lines: 141 – 143: How less permeable gut allow more LPS into the bloodstream?
Lines 149 – 151: What do you mean by disorders that don’t work properly? Disorders themselves are not functioning properly. Rephrase the sentence for scientific accuracy.
Line 154 -155: such Lactobacillus Rhamnosus – such as Lactobacillus Rhamnosus
Line 173-174: Elaborate “This growth of microbes happens at important times for the brain to mature.”
Lines 414-415: rephrase the sentence for clarity “This suggests that these bacteria generally assist keep the stomach working well.”
Line 520: Correct the phrasing “This produces SCFAs such acetate”
Lines 530-531: rephrase for grammatical correctness” There haven't been many research that looked at prebiotics alone in people with ASD”
Comments on the Quality of English Language
The authors should do a major revision to improve the academic/scientific writing style and address grammatical errors and inaccuracies in the review article. While some issues have been pointed out, it is recommended that the entire draft be thoroughly reviewed.
Author Response
Response to Reviewer 3 Comments
We appreciate very much the Reviewers’ interest in our work, for the helpful comments as well as the effort and time put into the review of this manuscript. All comments have been carefully considered and responded point by point.
Comment 1: " It would be beneficial to include information about variations in gut microbiota in ASD people with a focus on inter-individual variations.
Response 1: We thank the Reviewer for this comment. We have included information about variation in gut microbiota in ASD individuals. Specifically, in Section 5, we added a paragraph discussing how the gut microbiota composition in ASD can vary significantly between individuals due to multiple factors, including geography, diet, environment, genetic background and medication. This addition highlights that such variability may contribute to the heterogeneity of ASD presentations and treatment responses.
Comment 2: Line 269-270: provide information about the sample used for this transplantation “gut microbiota from human ASD donors”
Response 2: We thank the Reviewer for this comment. Ιn Section 10.4 of the revised manuscript, we have clarified the sample source by specifying that “FMT is the process of transferring stool (and the whole microbial population) from a healthy human donor to the GI tract of a recipient. It is a radical way to change the gut microbiota [111]. To address safety concerns, FMT human donor screening involves a detailed medical and lifestyle history, laboratory testing of blood and stool for infectious agents and quarantine measures. Repeat testing is performed for regular donors. These steps are consistent with recent gens [112]”(line 565-571). This addition ensures transparency regarding the origin of the transplanted microbiota and the safety measures applied.
Comment 3: Line 526: SCFA’s do not change how neurotransmitters are made. They may influence the levels of neurotransmitters either directly or indirectly. Rephrase the sentence for scientific accuracy.
Response 3: We thank the Reviewer for this important clarification. In the revised manuscript, we have rephrased the sentence to ensure scientific accuracy. The updated text now reads: “Additionally, certain probiotic strains may influence neurotransmitter activity.” (line 487)
Comment 4: It is suggested to include more information about different studies that have tested different types of gut microbiota-altering diets for ASD patients.
Response 4: We thank the Reviewer for the comment. In the revised manuscript, we have expanded the sections describing dietary interventions that alter the gut microbiota in ASD, including updated references to recent studies on gluten-free/casein-free, specific carbohydrate, low-FODMAP, and Mediterranean diets. These additions provide more comprehensive coverage of the different dietary approaches tested in ASD populations.
Comment 5: A more formal tone in writing is recommended. For example, instead of saying, “The brain releases corticotropin-releasing hormone (CRH), which tells the adrenal glands to make cortisol,” you can say, “stimulates the adrenal glands to release CRH.”
Response 5: We thank the Reviewer for the comment. The manuscript has been revised to ensure a more formal and scientifically precise style throughout.
Comment 6: Avoid using contractions like it’s,” “don’t,” “can’t,” “won’t, isn’t etc. in scientific writing. Thoroughly review the whole review article to address these issues.
Response 6: We thank the Reviewer for this observation. The entire manuscript has been carefully reviewed to ensure consistency with this standard.
Comment 7: This advice applies throughout the entire review article; authors are encouraged to carefully check for informal words and replace them with more formal language.
Response 7: We thank the Reviewer for this helpful advice. We have carefully revised the entire manuscript to replace all informal words and expressions, as suggested, with more formal scientific language, ensuring consistency in tone throughout the review.
Comment 8: The authors should do a major revision to improve the academic/scientific writing style and address grammatical errors and inaccuracies in the review article
Response 8: We sincerely thank the Reviewer for this valuable observation. Τhe manuscript has been thoroughly revised to improve academic and scientific writing style. We carefully corrected grammatical errors, refined sentence structures, and ensured greater accuracy and clarity throughout the text. Informal wording and contractions were also removed to maintain a formal academic tone.
Round 2
Reviewer 2 Report
Comments and Suggestions for Authors
The authors well replied to my previous comments
Comments on the Quality of English LanguageMinor editing is required
Author Response
Dear Reviewer,
Thank you very much for your valuable feedback. We would like to inform you that the manuscript has undergone professional language editing through the Authors Services, ensuring clarity, accuracy, and compliance with academic standards. We have carefully reviewed the text and confirm that it is ready for resubmission.
Sincerely,
Dr. Petropoulos Andreas
Reviewer 3 Report
Comments and Suggestions for Authors
I still see some informal contractions that need to be addressed.
Author Response
Dear reviewer, we are grateful for the time you have devoted to our submission and for the valuable observations. To further enhance the manuscript, we have engaged the Author Services of Nutrients to ensure the text is revised in accordance with a formal academic style. The necessary arrangements for payment have already been made, and the editing process will be completed shortly.
Please rest assured that we will carefully incorporate all suggestions regarding language and style so that the final version is fully consistent with the journal’s academic requirements.
With kind regards,
Dr. Andreas Petropoulos